



**Yields and molecular composition of gas phase and secondary**
**organic aerosol from the photooxidation of the volatile consumer**
**product benzyl alcohol: formation of highly oxygenated and**
**hydroxy nitroaromatic compounds**
Mohammed Jaoui[1], Kenneth S. Docherty[2], Michael Lewandowski[1], Tadeusz E. Kleindienst[1]
[1]Center for Environmental Measurement & Modeling, U.S. Environmental Protection Agency, Research Triangle Park,
NC, 27711, USA
[2]Jacobs Technology, Inc., Research Triangle Park, NC, 27709, USA
*Correspondence:* Mohammed Jaoui (Jaoui.mohammed@epa.gov)







**Abstract.** Recently, volatile chemical products (VCPs) have been increasingly recognized as important precursors for secondary organic aerosol (SOA) and ozone in urban areas. However, their atmospheric chemistry, physical transformation, and their impact on climate, environment, and human health remain poorly understood. Here, the yields and chemical composition at the molecular level of gas and particle phase products originating from the photooxidation of one of these VCPs, benzyl alcohol (BnOH), are reported. The SOA was generated in the presence of seed aerosol from nebulized ammonium sulfate solution in a 14.5 $m^3$ smog chamber operated in flow mode. More than 50 organic compounds containing nitrogen and/or up to seven oxygen atoms were identified by mass spectrometry. While a detailed non-targeted analysis has been made, our primary focus has been to examine highly oxygenated and nitro-aromatic compounds. The major components include ring-opening products with high oxygen to carbon ratio (e.g., malic acid, tartaric acids, arabic acid, trihydroxy-oxo-pentanoic acids, and pentaric acid), and ring-retaining products (e.g., benzaldehyde, benzoic acid, catechol, 3-nitrobenzyl alcohol, 4-nitrocatechol, 2-hydroxy-5-nitrobenzyl alcohol, 2-nitrophloroglucinol, 3,4-dihydroxy-5-nitrobenzyl alcohol). The presence of some of these products in the gas and particle phases simultaneously provides evidence of their gas/particle partitioning. These oxygenated oxidation products made dominant contributions to the SOA particle composition in both low and high NOx systems. Yields, organic mass to organic carbon ratio, and proposed reaction schemes for selected compounds are provided. The aerosol yield was 5.2% for BnOH/$H_2O_2$ at SOA concentration of 52.9 µg m$^{-3}$ and ranged between 1.7-8.1 % for BnOH/NOx at SOA concentration of 40.0-119.5 µg m$^{-3}$.

**Key words:** Benzyl alcohol, highly oxygenated compounds, Consumer products, VCPs, Silylation, Yield, Nitroaromatic compounds, SOA



## 1 Introduction

Modeling atmospheric organic aerosol (OA) using chemical transport models (CTMs) is complex, challenging, and often can lead to model-measurement discrepancies (Zhao et al., 2016). Applying CTMs to urban areas reveals that traditional VOCs including combustion-related processes cannot account for the observed OA mass, leaving a substantial fraction unresolved (Hayes et al., 2015). Recent studies suggest that this discrepancy is due in part to unaccounted, rapidly reacting SOA and ozone precursors from unknown sources (Hodzic et al, 2009; Hayes et al., 2015; McDonald et al., 2018; Akherati et al. 2019; Lu et al., 2020). Volatile chemical products (VCPs), such as personal care products, cleaning agents, coatings, adhesives, and pesticides have emerged as possible sources in urban areas (McDonald et al., 2018). Their emissions can be larger than those from usual sources, such as motor vehicles (Coggon et al., 2021). Laboratory, modeling, and field studies for VCPs have been conducted to assess their potential to affect ambient OA and ozone formation in urban and suburban locations (McDonald et al., 2018; Khare et al., 2018; Stockwell et al., 2021; Seltzer et al., 2021; Gkatzelis et al., 2021; Milani et al., 2021; Pennington et al., 2021; Coggon et al., 2021). The contribution of VCPs to ambient OA is not fully understood and only limited modeling studies have been reported (Mohr et al., 2015; Vlachou et al., 2018; Pennington et al., 2021; Qin et al., 2021; Seltzer et al., 2021). Additionally, few experimental and chamber studies of VCPs have been conducted with limited characterization of aerosol products (Wu and Johnston, 2016, 2017; Harrison and Well, 2012; Charan et al., 2020, 2021; Humes et al., 2022). For example, the analysis of SOA from the oxidation of cyclic methyl siloxanes (Wu and Johnston, 2016, 2017; Fu et al. 2020; Alton and Browne, 2020; Charan et al., 2021) and cyclic siloxanes (Janechek et al., 2019) has been conducted. Kinetic studies with limited products characterization have been reported for the oxidation of benzyl alcohol (BnOH) by hydroxyl radicals (Bernard et al., 2013; Wang, 2015; Harrison and Well, 2009, 2012). Recently, Humes et al. (2022) highlight the importance of oxygenated aromatic VCPs emission to generate urban SOA and oxygenated products in both gas and aerosol phases. Therefore, understanding the atmospheric chemistry of VCPs is important to assess their role in air quality and climate and to improve SOA chemistry in CTMs thereby allowing for better estimates in health studies and source apportionment.

The challenges associated with evaluating VCP impacts on urban OA can be addressed by identifying atmospheric VCP concentrations and SOA markers linking those VCP to ambient particulate matter (PM). Benzyl alcohol ($C_7H_8O$) is an important ring containing VCP used as an organic intermediate and a solvent in a wide range of applications (Antonelli et al. 2002). BnOH is emitted also from flowers and flowering trees (Do et al., 1969; Horvat et al., 1990; Larsen and Poll, 1990; Humpf and Scheier, 1991; Boatright et al., 2004; Vallat and Dorn, 2005; Orlova et al.; 2006) and found in indoor



air (Weschler, 2011). Gas kinetic studies of loss rates and product distributions have been conducted using flow tubes and
environmental chambers. Bernard et al. (2013) examined the rate and mechanisms of the OH + BnOH reaction. Similarly,
Harrison and Wells (2009, 2012) investigated the rate constants for the BnOH reaction with ozone, OH and $NO_3$ radicals.
Carter et al. (2005) conducted chamber experiments to assess ozone and PM formation from BnOH and related
compounds. Product studies from BnOH oxidation have focused mainly on gas phase (GP) products. Several carbonyl
products (benzaldehyde (BnAld), formaldehyde, glyoxal, butenedial, 4-oxopentanal, 3-hydroxy-2-propanaldehyde), and
benzyl nitrate, o-hydroxybenzyl alcohol, o-dihydroxy benzene were reported from the above studies. With respect to the
particle phase (PP), Charan et al. (2020) reported aerosol yields from BnOH oxidation together with a limited number of
SOA products. Finally, Wang (2015) conducted a theoretical study to elucidate the reaction mechanism of the oxidation
of BnOH with OH radicals.
In this study, we report a detailed non-targeted chemical analysis of GP and SOA products originated from the
photooxidation of BnOH in the presence and absence of oxides of nitrogen (NOx), with the aim to better understand the
chemical composition at the molecular level. Gas chromatography-mass spectrometry (GC-MS) and high-performance
liquid chromatography were used for the identification of a range of organic compounds including oxygenated
nitroaromatics and related compounds bearing up to seven oxygen atoms. Nitroaromatics are pollutants of concern due to
their toxicity, light-absorption properties, and relatively long residence times in the environment. Highly oxygenated
compounds can partition into pre-existing particles or be involved in new particle formation. Also, in the present study,
SOA and secondary organic carbon (SOC) yields were measured with the results compared to published data. A chemical
mechanism is then proposed to represent and account for selected gas- and aerosol-phase products observed in this study.

## 2 Experimental methods

All chemicals including N, O-*bis*(trimethylsilyl) trifluoroacetamide (BSTFA) derivatization reagent with 1%
trimethylchlorosilane (TMCS) as catalyst and benzyl alcohol (99%), 2-methyl-4-nitrophenol, L-(+)-tartaric acid, D-(-)-
tartaric acid, and meso-tartaric acid were purchased from Aldrich Chemical Co. (Milwaukee, WI) at the highest purity
available and were used without further purification. In addition to standards reported in our previous studies (Jaoui et al.,
2004; 2018), 3-nitrobenzyl alcohol, benzoic acid, and 4-nitrocatechol were purchased from Tokyo Chemical Industry
(OR, USA); while pentaric acid, 2,3-dihydroxy-4-methoxy-4-oxobutanoic acid, and arabic acid were obtained from
Aurum Pharmatech, LLC (NJ, USA).





**2.1 Chamber description and operation**

All experiments were conducted in a 14.5 m$^3$ fixed-volume chamber having TFE Teflon coated walls and maintained at a positive pressure of 0.1 Torr. The chamber operation, procedures, and instrumentation have been described previously (Kleindienst et al. 2006; 2009), and here just experiment-specific details are primarily included. A combination of fluorescent bulbs having radiation from 300-400 nm was used to photolyze NO$_2$. In the absence of NO$_x$, the radiation system was altered to include UV-313 sunlamps to adequately photolyze H$_2$O$_2$. The chamber was operated in steady-state (SS or flow) mode to provide continuously stable effluent concentrations. Under these conditions, reactants and products equilibrate with the chamber surfaces to minimize irreversible losses of gases and particles. The SS operation allows for extended sampling periods to improve the accuracy and precision of the measurements (Shilling et al., 2008). Temperature, relative humidity, and UV light intensity were measured continuously with an uncertainty of 5%. Pre-experiment and post-experiment procedures (see section 2.5 below) were routinely carried out before and after each experiment to minimize contamination in the chamber. The reactant generation system provided constant sources of zero air, reactants, water vapor, and ammonium sulfate (AS) seed aerosol. The reactant flow of gases (e.g., NO$_x$) into the chamber was regulated using mass flow controllers. BnOH was injected using a syringe pump, vaporized in a heated glass bulb, and injected with zero air. For experiments in the absence of NOx, a 50% aqueous solution of H$_2$O$_2$ was vaporized and injected using a second syringe pump, and photolyzed to produce OH radicals. Typical chamber AS concentrations were approximately 1 μg m$^{-3}$. Each SS experiment went through an initial transient period of 18-24 h until the reactant and product concentrations reached equilibrium.

**2.2 Gas-phase measurements**

A wide variety of instruments were used to measure the reactants and products. Nitric oxide (NO) and NOx were measured with a TECO (Franklin, MA) oxides of nitrogen analyzer with an upstream nylon filter to remove nitric acid produced from OH + NO$_2$. The NOx analyzer was calibrated with a NIST-traceable NO standard. Initial H$_2$O$_2$ concentrations were estimated by UV absorption using the ratio of the H$_2$O$_2$ to O$_3$ absorbances at 254 nm, as described by Kleindienst et al. (2009). Experiments in the absence of NOx were conducted dry to avoid aqueous loss of H$_2$O$_2$. BnOH concentrations in the inlet and within the chamber were measured semi-continuously using an SRI Model 8610C compact



gas chromatograph with flame ionization detector (GC-FID; SRI Instruments, Torrance CA). The purity of the BnOH
was verified with GC-MS analysis.

Low molecular weight carbonyls and dicarbonyls were quantified by derivatization using 2,4-dinitrophenylhydrazine

(Smith et al., 1989). Samples were collected at 0.5 L min$^{-1}$ for 25 min and derivatized in a 4 mL solution of acidified
DNPH and then heated for 40 min at 70ºC. A 15-component hydrazone standard (comprising formaldehyde-,
acetaldehyde-, acrolein, acetone-, propionaldehyde-, crotonaldehyde-, methacrolein-, butyraldehyde-, 2-butanone-,
BnAld-, glyoxal-, valeraldehyde-, m-tolualdehyde-, methylglyoxal-, and hexaldehyde; AccuStandard, Inc.) at a free
carbonyl concentration of 30 μg mL$^{-1}$ for each component was used for calibration. Separate dihydrazone standards of
glyoxal-DNPH and methylglyoxal-DNPH were also formulated. Carbonyls were separated using a Hewlett-Packard (HP)
1100 HPLC system having an Agilent Zorbax ODS 4.6 x 250 mm, 5-μm column maintained at 30ºC eluted with binary
acetonitrile-water gradient. A 10 μL injection volume was used for all standards and samples. Carbonyls were quantified
by UV absorption with a diode array detector set to 360 nm. Control and sample processing were managed with HP
ChemStation software. More highly oxidized gas-phase organic species were also collected with a 60-cm, 4-channel
XAD4-coated annular denuder for off-line analysis (Jaoui and Kamens, 2001). Once collected, the denuders were
extracted and analyzed according to the methodology described in section 2.4 below.

**2.3 Aerosol-phase: bulk parameter measurements**

Organic carbon (OC) was measured using a semi-continuous elemental carbon-organic carbon (EC-OC) instrument

(Sunset Laboratories, Tigard, OR) (Offenberg et al., 2007). The pumping system draws chamber effluent through a quartz
filter at a rate of 8 L min$^{-1}$ with carbon-strip denuder to remove gas-phase organics that might interfere with the
measurement. With a sample collection time of 0.5 h and an analysis time of 0.25 h, the duty cycle for the measurement
of OC was 0.75 h (Lewandowski et al., 2015). The aerosol volume, size distribution, and total number density were
measured using a scanning mobility particle sizer (SMPS), (Model 3071A, TSI, Inc., Shoreview, MN) and a condensation
particle counter (CPC) (Model 3010, TSI, Inc., Shoreview, MN). The SMPS operating conditions were as follows: sample
flow 0.2 L min$^{-1}$; sheath flow 2 L min$^{-1}$; size scan from 19 to 982 nm.

**2.4 Molecular characterization of GP and PP oxygenated organic products**





A non-targeted chemical analysis was conducted focusing mainly on species bearing hydroxy and carboxylic groups
(Jaoui et al., 2004, 2013, 2018). For each experiment, six 47-mm glass fiber (GF) filters were taken for 24 h at a flow rate
of 16.7 L min$^{-1}$. A second set of samples used an in-line 60-cm XAD-4 coated annular denuder (followed by a GF filter)
and analyzed for gas-phase organic products (Jaoui and Kamens, 2001). After collection, GF filters were extracted by
sonication with 5 mL methanol for 1 h, and denuders were extracted with 30 mL 1:1 dichloromethane/methanol mixture
(Jaoui and Kamens, 2001). Prior to extraction, denuders and GF filters were spiked with *cis*-ketopinic acid (KPA), trans-
p-menth-6-ene-2,8-diol (PMD), and d$_{50}$-tetracosane (TCS) as internal/recovery standards (IS/RS). Denuder extraction
solvents were rotary evaporated to ~1 mL and filtered using 0.45-µm PTFE syringe filters. A 2 µL portion of this extract
was analyzed by GC-MS (Jaoui and Kamens, 2001). The remaining denuder and filter extracts were evaporated to dryness
under a gentle stream of N$_2$ at room temperature using an N-Evap evaporation bath (Organomation Associates, Inc.,
Berlin, MA), then derivatized with BSTFA (Jaoui et al., 2004). This technique provides a sensitive method for measuring
low levels of highly oxidized organic compounds, including semi- and intermediate-volatile compounds in the GP and
PP.
The GC-MS analysis was conducted on an Agilent GC (7890B) coupled with a quadrupole mass spectrometer
(5977B). The injector, heated to 270 °C, was operated in splitless mode. Compounds were separated on a 60-m-long, 0.25-
mm-i.d. RTx-5MS column (Restek, Inc., Bellefonte, PA) with a 0.25-µm film thickness. The GC oven temperature was
initiated at 84 °C, held for 1 min, then increased at 8 °C min$^{-1}$ to 200 °C, followed by a 2-min hold, then an increase at 10
°C min$^{-1}$ to 300 °C and a 15-min hold.  The ion source, ion trap, and interface temperatures were 200, 200, and 300 °C,
respectively.  Mass spectra were collected in both the methane-chemical (CI) and electron ionization (EI) modes.

**2.5 Experimental and quality control procedures**
Before each experiment started, the chamber was flushed with zero air for 24 h. After the reactants reached
equilibrium concentrations, the background was characterized using all instruments to check for artifacts including
background GP and PP species. Background chamber air was also characterized using off-line analysis of denuder and/or
filters as described above. Previous studies show that BnAld and to a lesser extent benzoic acid, benzyl benzoate and
dibenzyl ether present either as impurity, or as decomposition products upon BnOH exposure to air at room temperature
or sonication (Urakami et al., 2000; Ferri et al., 2006; Abend et al., 2004). Here we investigated the effect of chamber air,
sonication, and BSTFA derivatization on BnOH artifacts as described in the supplementary information (SI) in section 1.





A small amount of BnAld impurity was detected using the direct injection (DI) method and estimated to be <0.1% in the
purchased solution. When BnOH was exposed to clean air in the chamber in the absence of light, sonication, and/or
BSTFA derivatization, our results show additional low level of BnOH conversion to BnAld and benzoic acid using DI
and BSTFA methods, which is similar to the findings of Abend et al. (2004), Urakami et al. (2000), and Ferri et al. (2006).
Additional results and descriptions are provided in section S1 (SI).

Experiments were initiated by turning on the lights and allowing the irradiated chamber effluent to reach SS

conditions over a 24-h period which permits active sampling by the on-line instruments and the collection of denuder and
filter samples for subsequent off-line analysis. For organic intermediates wall losses are typically not an issue due to
reactions being conducted within a Teflon chamber. This potential issue is mitigated further from operating the chamber
in a SS mode where compound loss and re-evaporation quickly comes to equilibrium. Short lifetimes of radical
intermediates with other gas-phase constituents also render a negligible wall-loss. The stability of BnOH in the chamber
was investigated and BnOH was found to be highly stable with results given in the SI (section S1). Denuders and GF filter
samples were also analyzed to probe reproducibility of the analytical technique. The analysis showed consistent results.

Gas and particle samples from BnOH photooxidation are dominated by oxygenated species, several not having

authentic standards, and thus a portion of each sample was derivatized. Initially, we eliminated peaks detected in blank
and background samples. For compounds having standards, comparisons were made between the retention times and mass
spectra (CI and/or EI mode) of the chamber-derived peaks and those of the standards.  For compounds not having
standards, individual peak identifications were associated with a product peak only if its retention time and mass spectrum
was consistent with the fragmentation pattern of the BSTFA-derivatized compound. All recorded spectra in this study
were compared with those derived from reference standards, the literature, the NIST library, and an archive of mass
spectra from product compounds determined in our laboratory over the past twenty years.

**3 Results and discussion**

The initial conditions of the experiments conducted in this study are summarized in Table 1. Three NOx experiments

were carried-out with initial BnOH ranging from $0.36 – 0.72$ ppm and NO from $0.096 – 0.19$ ppm. One experiment without
$NO_X$ was conducted with initial $H_2O_2$ and BnOH levels of 3.0 and 0.32 ppm, respectively. NOx experiments were
conducted at ~30% RH, and the $H_2O_2$ experiment at < 4% RH to minimize $H_2O_2$ uptake onto chamber surfaces. Chamber



temperatures were set to 25 °C. Each experiment was conducted for up to five days for samples requiring substantial
masses or extended collection times and frequencies.
Steady state concentrations of NO, BnOH, $O_3$, and NOy for the four experiments are given in Table 2. The reacted
BnOH and NO were calculated from the difference between the initial and steady-state concentrations. For NOx
experiments, the range of reacted BnOH concentrations was 0.22 – 0.34 ppm having a reproducibility of 20–30%. Under
these conditions, steady state concentrations of NOy, and $O_3$ were in the range of 0.08 – 0.16 and 0.011 – 0.15 ppm,
respectively. With NO present at steady-state, peroxy-peroxy ($RO_2$–$RO_2$) reactions were minimized. A constant aerosol
source was maintained for initial conditions given in Table 1. The major aerosol parameters measured (SOA, SOC, and
OM/OC) are given in Table 3. SOC uncertainties were taken from the reproducibility of the semi-continuous measurement
and typically better than 10% for a single run. For the organic mass (OM), the uncertainties are determined from the
reproducibility of side-by-side filter measurements which are typically better than 5%. An estimate of the systematic
errors due to minor changes in reactant concentrations, minor variations in chamber temperature, and similar factors bring
the total uncertainty to between 15-25% for these parameters (Kleindienst et al., 2009). SOA/SOC values were then
determined from the corrected data and given in Table 3. For experiments in the presence of NOx, SOA/SOC values
ranged from 1.7-2.0. Similarly, in the absence of NOx, the measured SOA/SOC value was 2.1.

**3.1 Secondary organic aerosol and secondary organic carbon yields**
Secondary organic aerosol yield ($Y_{SOA}$) and secondary organic carbon yield ($Y_{SOC}$) were calculated from the
following respective relationships $Y_{SOA} = SOA/\Delta HC$ (1); $Y_{SOC} = SOC/\Delta HC_C$ (2) where SOA is the corrected organic
aerosol mass concentration originated from filter measurements (6 filters) and $\Delta HC$ is the reacted BnOH concentration.
SOC is the organic carbon concentration found in Table 3 and $\Delta HC_C$ is the reacted BnOH carbon concentration. SOA and
SOC were corrected for wall loss to the chamber which had previously been determined for organic aerosol to be 0.067
$h^{-1}$ (Kleindienst et al. 2012). Uncertainties in the yield come from the experimental uncertainties in SOA and SOC
production and the reacted BnOH concentrations. The uncertainty in the reacted BnOH results from the reproducibility
of the initial and steady-state values and is estimated to range from 20 - 30% given the low volatility of BnOH and
challenges for introducing oxygenated species into the chamber in a consistent manner. Such challenges are also present
in a batch mode experiment given the difficulty to determine BnOH time profiles given its volatility and high reactivity
toward oxidants (Shilling et al., 2008; Kroll et al., 2007). Similar findings have been reported for sesquiterpenes oxidation



(Jaoui et al., 2013). As a result, aerosol yields of higher accuracy are often reported to be associated with steady state as
opposed to batch mode experiments (Shilling et al., 2008). Moreover, in this work we explored the possibility of BnOH
being taken up by ammonium sulfate (AS) seed aerosol prior to start of the irradiation or by SOA after it is initiated. This
test was conducted using GC-MS analysis of derivatized (BSTFA) and underivatized denuder and GF filter extracts
collected before and after the reaction starts (SI; Section S1). Under the experimental conditions used in this study, BnOH
was undetected in AS and SOA, thus limiting any participation in particle chemistry that may occur.

The production of aerosol, and thus the yield, were found to be highly sensitive to the precise initial conditions

(Tables 1, 3). Yields for the four experiments are shown in Table 3. $Y_{SOA}$ values were determined for SOA concentrations
from 39.6 - 119.5 $\mu g\ m^{-3}$ and ranged between 3.6 and 8.1%. Similarly, $Y_{SOC}$ was measured for SOC concentrations from
23.2 – 58.9 $\mu gC\ m^{-3}$ and found to range between 2.7 and 5.1%. In the absence of $NO_X$, SOA and SOC yields were 5.2
and 3.1 % measured for SOA and SOC concentrations of 52.9 $\mu g\ m^{-3}$ and 24.8 $\mu gC\ m^{-3}$, respectively. For the two systems
at similar SOA concentrations ER890 and ER892, the SOA yield was higher for the experiment with $NO_X$. This may
result from the reaction of BnOH with $NO_x$ which tends to produce high levels of BnAld (Table 4), which may undergo
secondary reactions leading to additional SOA formation (see section 3.3). As expected, the data in Table 3 indicate that
$Y_{SOA}$ and $Y_{SOC}$ are lower at the lower SOA and SOC concentrations, respectively.

These SOA and SOC yields can be compared with other studies. Recently, Charan et al. (2020) reported SOA yields

for the photooxidation of BnOH in the presence of $NO_X$ with the initial OH coming from the photolysis of $H_2O_2$. Their
chamber was operated in a batch mode and SOA yields approaching unity were reported. By contrast, three additional
studies reported much lower SOA yields of 0.09, 0.3, and 0.41 from McDonald et al. (2018), Carter et al. (2005), and Li
et al. (2018), respectively. The yield reported by McDonald et al. (2018) was based on a multi-generation oxidation model;
that of Carter et al. (2005) was estimated as described in the original report, and that of Li et al. (2018) was based on
measurements in the presence of $NO_x$ and a surrogate urban hydrocarbon. The results of our study are much closer in
value to McDonald et al. (2018). The study by Charan et al. (2020) suggests that conditions can be found where BnOH
SOA yields are substantially greater than that found in this study and those previously reported. The major differences
between the Charan et al. study and the present work were the chamber-mode operation, the seed aerosol type and levels,
and the mix of oxidants used. While it can be difficult to compare the yields from the two studies some comments can be
made. (1) As noted, the Charan et al. yields result from conventional batch mode irradiations of BnOH, $H_2O_2$ and $NO_X$.
(2) SOA levels were measured using an SMPS which measures aerosol volume which is then converted to aerosol mass


M$^{+\cdot}$ + 29, and M$^{+\cdot}$ + 1; fragment ions include $m/z$ M$^{+\cdot}$ – 15, M$^{+\cdot}$ – 73, M$^{+\cdot}$ – 89, M$^{+\cdot}$ – 117, M$^{+\cdot}$ – 105, M$^{+\cdot}$ – 133, and/or
M$^{+\cdot}$ – 207. The approach used for the identification is as follows: peaks detected in blank and background chamber samples
were eliminated first. A peak was associated with a reaction product only if its corresponding mass spectrum was
consistent with the fragmentation pattern of the BSTFA derivatization reagent. All recorded spectra were compared with
spectra derived from various reference compounds, authentic standard, NIST library, the PubChem website
(pubchem.ncbi.nlm.nih.gov), and/or by MS assignment. While the off-line technique is an integrated technique that
requires long sampling times, it does provide a sensitive method for products identification at the molecular level as well
as measuring low concentrations of highly oxidized organic compounds, and semivolatile compounds in the GP. Thus,
products found by this collection technique could be informative for possible precursors for the types of compounds that
may form in the PP. In the following discussion, data are first presented to support tentative identifications of oxidation
products in the GP and PP.
**Gas phase products.** GP measurements were made of major carbonyl products formed during the photooxidation of
BnOH including formaldehyde, acetaldehyde, acetone, methacrolein, 2-butanone, BnAld, glyoxal, and methylglyoxal.
Steady-state concentrations are given in Table 4. Under the conditions shown in Tables 1 and 2, high concentrations were
observed for BnAld and glyoxal, and to a lesser extent formaldehyde in experiments with NOx, and high concentrations
of formaldehyde, acetaldehyde, and to a lesser extent BnAld and glyoxal in the experiment without NOx. BnAld level
was a factor of ~5 higher in NOx experiments compared to H$_2$O$_2$ experiments, and formaldehyde a factor of ~36 lower.
Glyoxal and methylglyoxal concentrations largely were similar in both NOx and H$_2$O$_2$ experiments. The formation of
BnAld, glyoxal, and formaldehyde as major products (Table 4) have already been reported from the oxidation of BnOH
with yields of 25 (± 5), 20 (± 2), and 3.0% (± 0.2), respectively (Bernard et al., 2013; Harrison and Wells, 2012).
GP samples were also collected on five-channel annular denuders. Each denuder sample was extracted and analyzed
directly with GC-MS without derivatization. The remaining extract was silylated, and GC-MS analyzed qualitatively.
Typical total ion chromatograms (TIC) of GP products detected and identified in this study are shown in Figure 1. Figure
1 shows portions of three TIC in +EI of GP samples taken from experiments ER889 at steady state (underivatized: Figure
1a), ER892 (underivatized: Figure 1b), and ER889 (silylated derivatives: Figure 1c). Peaks assigned in Figure 1 were
identified either by comparison with an authentic standard or by MS assignment. For clarity, only the main products are
shown, although several peaks could not be structurally identified. SOA generated from BnOH photooxidation is





dominated by oxygenated ring-opening products (see below). However, ring-retaining products were among the main
products observed in the GP including semivolatile organic compounds (SVOCs) (Figure 1c). Chromatograms associated
with the underivatized samples (Figure 1a, b) were used mainly to identify BnOH and BnAld in the system, although
several additional peaks absent in the background chromatogram were observed. At steady state, BnOH was not reacted
completely as it was detected in both systems only in the GP using both DI and BSTFA methods (Figure 1). BnAld was
detected in both systems in the gas and particle phases, although it was not present with BSTFA method because of the
absence of OH or COOH groups. Figure S3 shows EI mass spectra of BnAld identified using authentic standard, and
those associated with three peaks eluting at 11.3, 12.0, and 12.8 min. Although no structural information could be
associated with these three peaks, molecular weights of 152, 152, 138 Da (all derivatized and underivatized masses are
Dalton (Da) but are not designated as such hereafter), were tentatively obtained.

Select GP products containing OH groups identified in the present study are summarized in Table 5. Table 5 contains

proposed structures, molecular weights of the silylated derivatives ($MW_{BSTFA}$) and underivatized compounds (MW),
formula, and the 5 most intense ions associated with BSTFA derivatives in EI mode. Table 5 shows if GP products are
detected also in the PP. Figure S4 shows EI mass spectra associated with selected peaks observed in Figure 1c, including
BnOH-1TMS, benzoic acid-1TMS, catechol-2TMS, and 2-hydroxybenzyl alcohol-2TMS. 2-Hydroxybenzyl alcohol-
2TMS (2OHBnOH) peak eluted at 21.4 min was one of the largest peaks detected in the chromatogram in Figure 3c. The
2OHBnOH-2TMS EI mass spectrum (Figure S4, bottom) shows strong characteristic fragments ions at $m/z$ 73, 179 ($M^{+\cdot}$
- 89), 253 ($M^{+\cdot}$ - 15), 268 ($M^{+\cdot}$), and its corresponding CI mass spectrum shows ions at m/z 253 ($M^{+\cdot}$ - 15), 179 ($M^{+\cdot}$ - 89)
and weak adducts at $M^{+\cdot} + 1$, $M^{+\cdot} + 29$, and $M^{+\cdot} + 41$ that are consistent with the presence of two (-OH) groups, indicating
a BSTFA derivatized molecular weight of 268 Da. Bernard et al (2013) have identified 2OHBnOH and catechol in the
GP of the reaction of BnOH and OH radicals. In our study, catechol was observed only in the $H_2O_2$ system in the PP.
Additional peaks were observed, which their mass spectra are consistent with products bearing OH and/or COOH groups,
however, their structural identification could not be obtained due to lack of authentic standards and the complexity of the
interpretation of their mass spectra. BnAld was reported to undergo secondary reactions (Sankar et al., 2014) and may
play an important role as precursor for some oxygenated species observed in this study.
**Particle phase products.** One of the advantages of conducting experiments in SS mode is collecting sufficient gas and
aerosol masses on denuders and filters for qualitative and quantitative offline analysis. In this study, aerosol collected on



GF filters were solvent extracted, with the resulting extracts subjected to BSTFA derivatization followed by GC-MS
analysis. SOA generated from both NOx and $H_2O_2$ systems was dominated by oxygenated organic compounds, for which
mass spectra for more than 50 species have been recorded. These species may have undergone several generations of
atmospheric oxidation. Several individual large peaks have been detected in addition to a significant number of small
peaks as shown in Figure 2. Figure 2 shows portions between 9 and 28 min of the TIC chromatograms of the silylated
derivatives of the aerosol extracts associated with BnOH/NOx (top) and BnOH/$H_2O_2$ (bottom). The portion after 28 min
is discussed in the next section. The chromatograms in Figure 2 can be directly compared because the chamber air sampled
and the amount of extract analyzed for each system were the same. This evaluation revealed that more than 70% of peaks
eluted from each system are identical, suggesting similar chemistry is involved in BnOH reaction products formed in the
presence and absence of NOx. In addition, a series of peaks dominated by fragments with odd *m/z* were observed only in
BnOH/NOx and their mass spectra were associated with nitrogen containing compounds as discussed in the NACs section
below. This suggests that the composition of a portion of SOA produced in the presence of NOx is different than that
formed in the absence of NOx, which can be clearly illustrated by the filters and extracts color shown in Figure 2 (bottom).
Consistent with the presence of nitroaromatics, filter F2 and methanol extract (E2) has lost most of the color seen in F1
and E1. The presence of NOx in the system produced material (filter F1) of a deep brown color. Most species structurally
identified in this study have not been reported in the literature, and mass spectra associated with several peaks are provided
either in the main manuscript or in the SI. Additional reaction products (e.g., oligomers, organonitrates) might have been
present in the SOA but could not be detected based on the analytical techniques used in this study. Note that formulae, in
particular chemical structure, could not be obtained for several peaks recorded in this study due to challenges interpreting
their mass spectra. A set of compounds identified and detected before 28 min in the present study are summarized in
Table 5.

Ring retaining products (e.g., 2-hydroxy benzyl alcohol, benzoic acid, 4-hydroxy benzoic acid, and catechol) were

detected in the PP in both systems, while catechol was detected only in the absence of NOx. As noted above, some ring-
retaining products were detected also in the GP as shown in Table 5. Salicylaldehyde and 3-hydroxybenzaldehyde were
present only in the GP. These two hydroxy-aldehydes may undergo additional secondary reactions leading to some ring-
opening products observed in this study. Representative EI mass spectra of the TMS-derivatives associated with four
compounds are shown in figure 3 including benzoic acid, benzene-1,2-diol (catechol), 4-hydroxybenzoic acid, and 2-
hydroxybenzyl alcohol. Additional EI and CI mass spectra are shown in figures S4 and S5 in the SI. The EI mass spectrum



of the BSTFA derivative of 2-hydroxybenzyl alcohol displayed in figure 3 shows abundant fragment ions at $m/z$ 73, 147,
267 ($M^{+\cdot}$), 253 ($M^{+\cdot} - 15$), and 179 ($M^{+\cdot} - 89$), and weak ions at $m/z$ 91, 223 and 163. The corresponding CI mass spectrum
displayed in figure S4d shows abundant fragment ions at $m/z$ 268 ($M^{+\cdot}$), 253 ($M^{+\cdot} - 15$), and 179 ($M^{+\cdot} - 89$) and adduct
ions at $m/z$ 293 ($M^{+\cdot} + 29$) and 309 ($M^{+\cdot} + 41$). This fragmentation pattern is consistent with the presence of a compound
with two hydroxyl groups and a benzene ring ($m/z$ 91) having molecular weight 268 for the BSTFA derivative, and MW
124 for its underivatized form. Similarly, the BSTFA EI mass spectrum of 4-hydroxybenzoic acid (Figure 3c) shows
characteristic fragment ions at m/z 73, 193 ($M^{+\cdot} - 89$), 223 ($M^{+\cdot} - 60$), 267 ($M^{+\cdot} - 15$) and 282 ($M^{+\cdot}$), and its CI mass
spectrum fragment ions at m/z 73, 193, 67 and adducts at 283, and 311. Again, these fragments and adducts are consistent
with the presence of two (-OH) groups and a molecular weight of the derivatized compound of 282 and 138 for the
underivatized compound. The presence of a peak at m/z 153 ($M^{+\cdot} - 117$) is consistent with a compound bearing an organic
acid group. The EI mass spectra recorded in this study for 2-hydroxybenzyl alcohol and 4-hydroxybenzoic acid are
identical to the reference NIST spectrum (webook.nist.gov). Figure S5 shows EI mass spectra associated with four peaks
eluted at 12.86, 15.58, 16.24, and 19.78 min consistent with the fragmentation pattern of BSTFA derivatives, although
their structures could not be obtained.
**Highly oxygenated compounds (HOCs).** Recent studies show that highly oxygenated compounds (e.g., HOMs) play an
important role in understanding SOA formation (Berndt et al. 2016, Jaoui et al., 2019; 2021 and references therein, Piletic
and Kleindienst, 2022). These compounds may result from several generations of atmospheric oxidation. In this study,
several ring-opening products eluted late in the chromatograms (RT > 25 min), with a relatively high O:C ratio of > 1.3
likely contributes to their condensation in the PP, were detected. Three groups of these oxidation products were detected
in the PP in both systems. Figure 4 shows the portion between 25 and 34 min of selected GC-MS extracted-ion
chromatograms where these groups (color coded) elute and uses the selected ions m/z 423, 437, and 525 (merged in one
chromatogram) to best illustrate them: (a) BnOH/NOx; (b) BnOH/H2O2; (c) chamber background. Groups 1, 2, and 3
consist of three (green), eight (blue) and four (red) peaks, respectively, and are completely absent from the background
chromatogram (figure 4c). Results from a comprehensive interpretation of EI and CI mass spectra associated with peaks
shown in figure 4 enabled the identification of several isomers associated with each group. Figure 5 displays three EI
mass spectra associated with each group main peak, along with proposed structure and chemical formulae. Table 6 gives
the major highly oxygenated compounds identified in this research, including the main peaks from each of these groups,
in the order of their underivatized molecular weight. Table 6 gives the chemical formulas, O:C mass ratio, the five most



abundant ions associated with each TMS derivative in methane-CI and EI modes, the molecular weights of the
underivatized (MW) and TMS-derivatized compounds (MW$_{BSTFA}$), and the proposed chemical structures of the
compounds.
Group 1 consists of *meso*-tartaric acid (*m*TA) (Rt 26.04 min), and *L*-/*D*-tartaric acids (*l*TA/*d*TA) (Rt 27.66 min)
identified based on authentic standards. The mass spectra of BSTFA derivatives of *l*TA and *d*TA standards (Figure S6,
SI) are very similar (eluting at the same time) and are only slightly different from the *m*TA (Figure S6: SI); however,
*l*TA/*d*TA and *m*TA elute at two different retention times (Figure 4, S6). The peak associated with *m*TA, and *l*TA/*d*TA are
among the largest peak observed in this portion of the chromatograms. Note, *l*TA isomer is the most abundant tartaric
acid present in nature (DeBolt et al., 2006). The fragments and adducts observed for the peak eluting at 25.19 min are
similar to those of mTA and d-/lTA and are consistent with the presence of four OH groups, a MW of 452 for the
derivatized compound and 164 for the underivatized compound, and a $C_4H_6O_6$ chemical formula. Tartaric acid has been
reported in ambient aerosol (Rohrl and Lammel, 2002; Gowda et al., 2016) and in chamber 1,3-butadiene SOA (Jaoui et
al., 2014). Recent studies suggest that tartaric acid and other hydroxy carboxylic acids undergo heterogeneous OH reaction
in aqueous solution, with the presence and position of OH group(s) playing an important role in fragmentation and
functionalization of organic aerosol (Cheng et al., 2016).
Group 2 consists of eight peaks (figure 4: blue) eluting between 28.5 and 31.5 min. The EI and CI mass spectra
associated with each peak display similar fragment and adduct ions across the range of 50 to 600 Da. The interpretation
of these mass spectra allows us to infer the molecular weight (MW) of the underivatized compounds as 164 and MW$_{BSTFA}$
of 452 for the TMS derivatives. The BSTFA CI mass spectrum of the peak eluted at 29.48 (largest peak) shows
characteristic fragment ions at m/z 73, 437 [M$^{+\cdot}$ -15], 363 [M$^{+\cdot}$ - 89], and 305 [M$^{+\cdot}$ - 105], and an adduct at 453 [M$^{+\cdot}$ + 1],
[M$^{+\cdot}$ + 29], and 493 [M$^{+\cdot}$ + 41]. These fragments and adducts are consistent with the presence of four OH groups and
an MW of 452 for the derivatized compound and 164 for the underivatized compound. The presence of peaks at m/z 347
[M$^{+\cdot}$ - 105], and 335 (M$^{+\cdot}$ – 117) are consistent with a compound bearing alcoholic and carboxylic OH groups
simultaneously. This mass spectrum is similar to the one from methyltartaric acid reported previously from isoprene
oxidation by our group (Jaoui et al., 2019). The silylated methyltartaric acid mass spectrum (Jaoui et al., 2019) and mass
spectra associated with group 2 are only slightly different, however, they elute at different retention times. The peaks have
been tentatively identified as isomers of trihydroxy-oxo-pentanoic acid, with the structure of 4-oxo-D-arabonic acid
isomer shown in Table 6.





Group 3 consists of four peaks eluting between 32.5 and 34 min (figure 4: red). The EI and CI mass spectra associated
with each peak display similar fragment and adduct ions across the range of 50 to 600 Da. As a descriptive example, an
EI mass spectrum is shown in Figure 5 for peak eluted at 33.1 min. A comprehensive interpretation of EI and CI mass
spectra associated with group 3 peaks (Figures 4, 5), allows us to infer the molecular weight (MW) of the underivatized
compounds as 180, and $MW_{BSTFA}$ of 540 for the TMS derivatives, with a chemical formulae $C_5H_8O_7$. The compounds
corresponding to these four peaks were identified as isomers of $C_5$-trihydroxydicarboxylic acids. This identification is
tentative due to the absence of authentic standards, except for peak eluting at 33.47, which was identified as pentaric acid
(Table 6) based on authentic standard. The spectra of BSTFA derivatives of the remaining three red peaks are only slightly
different from the pentaric acid spectrum (Figure 6); however, they elute at different retention times. The EI mass spectra
are also similar to those reported in the literature for a set of C5-aldaric acids-TMS derivatives including xylaric, arabinaric
and ribaric acids (Hinton et al., 2008; https://pubchem.ncbi.nlm.nih.gov). Figure 6 shows the structure of pentaric acid
and its four isomers (a), the spectra associated with BSTFA derivative of pentaric acid observed in BnOH SOA (b: EI
mode), (c: CI-CH$_4$ mode), and standard (d: EI mode). Figure 6 also shows the structure of the main fragments observed
in BSTFA derivative of pentaric acid in EI mode including m/z at 540, 525, 407, 292, 147, and 73 Da. Pentaric acid and
its isomers (aldaric acids) are reported to be formed from the oxidation of aldopentose (Hinton, 2008; Derrien et al., 2018),
but no evidence has been provided for its presence in SOA samples. In the present study, we successfully identified aldaric
acids from the oxidation of BnOH in SOA samples.

**Nitroaromatic compounds (NACs).** NACs of secondary origin are a possible contributor to urban OA and not only
adversely affect human health and the environment but impact the aerosol optical properties and the atmospheric radiation
balance. By understanding the sources of NACs in ambient particles and their chemical identities, we can evaluate their
impact on the climate, environment, and human health. Recently, the analytical capabilities associated with BSTFA
derivatization have been extended to NACs bearing hydroxyl and carboxylic acid groups (Jaoui et al., 2018). Mass spectra
of most silylated NACs, especially methane-CI, are highly specific, reproducible, and produce characteristic fragments
useful in determining structural information and molecular weight, when authentic standards are not available (Jaoui et
al., 2018). In this study, a detailed analysis of mass spectra associated with peaks in chromatograms Figure 1c (GP) and
Figure 2 top (PP) reveals the presence of several peaks presenting similar fragmentation patterns as those reported by
Jaoui et al. (2018) for species bearing hydroxyl, carboxylic, nitro groups, and benzene ring. Figure 7 shows the portion



between 23 and 42 min of two +EI extracted ion chromatograms for the BSTFA derivatives at $m/z$ 210, 165 (IS), 299 (IS),
300, 298, 372, 388 (merged in one chromatogram) associated with BnOH/NOx (top) and BnOH/H$_2$O$_2$ (bottom). The EI
and/or CI mass spectra of selected nitroaromatic standards can be found in Jaoui et al. (2018), and additional representative
subset of the derivatives are displayed in Figures S7 (SI). For clarity, figure 7 inset shows an expanded portion of the top
chromatogram between 26.7 – 28 min.  Table 7 contains proposed identification of NACs detected in this study, along
with molecular weights, formulae, main 5 intense ions associated with CI and EI mass spectra of the derivatives, proposed
structure, and the GP to PP peak area ratio.
NACs with the highest confidence assignment are those identified by comparing their retention times, EI, and CI
mass spectra with those of reference standards, and NACs with low levels of confidence are those (1) that have been
identified previously in ambient PM or in smog chamber studies, (2) their EI mass spectra exist in the literature, or (3)
their molecular weights and numbers of OH, COOH, and NO$_2$ groups are simply consistent with the CI and EI mass
spectrum (Jaoui et al., 2018). A total of fourteen peaks associated with NACs were detected in this study. 3-Nitrobenzyl
alcohol, 4-nitrocatechol, 2-hydroxy-5-nitro benzyl alcohol, and 2-nitrophloroglucinol were identified based on authentic
standards. Three peaks eluted at 33.76, 34.70, and 34.76 having similar mass spectra as 2-nitrophloroglucinol (main peak)
eluted at 35.62 min were detected. They were tentatively associated with homologous series of 2-nitrophloroglucinol
including 3-nitrobenzene-1,2,4-triol, 5-nitropyrogallol, and 4-nitro-1,2,3-benzenetriol (not shown in Table 7). Similarly,
three additional peaks having similar mass spectra as 2-hydroxy-5-nitrobenzyl alcohol were observed and were tentatively
associated with homologous series of 2-hydroxy-5-nitrobenzyl alcohol including 4-hydroxy-2-nitrobenzyl alcohol. The
EIC in figure 7 (top) includes a series of four peaks observed only in the PP eluting at 35.94, 36.60, 38.18 min, whose
mass spectra were consistent with the presence of molecular weight 185 and 401 for the underivatized and derivatized
compounds, respectively. Based on similarity of their mass spectra, they were tentatively identified as structural
homologue of 3,4-dihydroxy-5-nitrobenzyl alcohol (Table 7) with C$_7$H$_7$NO$_5$ formulae. As can be seen in figure 7 (bottom),
NACs peaks were not detected in BnOH/H$_2$O$_2$ SOA extract, consistent with the formation of NACs in the presence of
NOx. All NACs were detected in both GPs and PP (Table 7), except 2-nitrophloroglucinol and 3,4-dihydroxy-5-
nitrobenzyl alcohol and their isomers were observed only in the PP consistent with their low volatility. This result suggests
that NACs may be formed in the GP, and partition to the PP for those with low volatility, although PP reactions may
occur as suggested by Charan et al. (2020) who analyzed only PP. 4-Nitrocatechol and 2-nitrophloroglucinol were among
the largest NAC peaks observed in our study (Figure 7). All three experiments conducted in this study were analyzed for



NACs to probe reproducibility of the BSTFA method and showed consistent results. 2-Nitrophloroglycinol, 4-
nitrocatechol and other NACs has been reported in PM collected in Pico Mountain Observatory, Pico Island in the Azores
archipelago by Ikemori et al., (2019). A series of NACs have been reported recently by Charan et al. (2020) in BnOH
SOA using off-line UPLC/ESIQ-ToFMS (ultra-high-performance liquid chromatography electrospray ionization
quadruple time of flight mass spectrometry), and the structure assigned to formulas obtained from MassLynk software
was based on expected oxidation products and MS/MS analysis. These observations support the identification of NACs
reported in this study. 4-Nitrophenol was reported in the GP by Bernard et al., (2013) at low yield and by Charan et al.
2020 in SOA from the OH radical oxidation of BnOH but was not detected either in the GP or the PP in this study.

**3.3. Mechanism of product formation**

Based on known GP reactions for aromatic compounds, a schematic representation for the reaction of BnOH with

OH is presented in schemes 1-3. It is developed to understand the chemistry leading to the main GP and PP products
identified experimentally in this study including HOCs and NACs. These schemes incorporate the latest experimental,
quantum and kinetic developments of the fate of peroxy/alkoxy benzoyl radicals including autooxidation (Wang, 2015;
Sankar et al. 2014, and Namysl et al. 2020). The lines shown in these schemes are either one step or multistep pathways.
Rate constants at room temperature of BnOH with OH radical, $O_3$, and $NO_3$ radical of 2.8 $10^{-11}$, 6 x $10^{-19}$ (upper limit),
and 4.0 x $10^{-15}$ $cm^3$ molecule$^{-1}$ s$^{-1}$, respectively, have been reported in the literature (Harrison and Wells, 2009, 2012;
Bernard et al., 2013). This suggests that the day-time oxidation of BnOH will be mainly initiated by OH radicals. The
reaction for $O_3$ and $NO_3$ radical are not included in schemes 1-3, although they are expected to be formed as minor
products in our systems.

The reaction of BnOH with OH radicals is initiated primarily by H atom abstraction from the external $CH_2$ group

leading to BnAld, and OH addition to the aromatic *ipso* (C1) and *ortho* (C2 or C6) positions to form two alkyldihydroxy
adducts R1 and R2 (scheme 1). The OH addition to the *para* (C3, C5) and *meta* (C4) position was reported to be not
favourable based on theoretical study of Wang, (2015). Scheme 1 shows mechanistic pathways leading to the formation



**Scheme 1.** Initial reaction pathways proposed to produce selected products detected in this study (blue color) in the gas

or PP (Table 5). R1-26OO and R2-13OO intermediates undergo further reactions leading to ring-opening products as

shown in scheme 3.



of stable products (blue) including BnAld, 4-hydroxybenzyl alcohol, 2-hydroxybenzyl alcohol, phenol, formaldehyde,
and catechol. The initial branching ratios shown in scheme 1 are those reported by Wang (2015), obtained by combining
quantum chemistry calculations and experimental work from the literature. BnAld was observed in this study in the
presence and absence of NOx, and its secondary chemistry may lead in part to oxygenated compounds observed in this
study (Bernard et al., 2013). Due to large number of possible intermediates formed (Wang, 2015), only selected pathways
energetically favourable leading to some products observed in this study are considered. We refer the readers to Wang
(2015) paper for an in-depth theoretical analysis of mechanistic pathways leading to the formation of selected reaction
products. The adduct R1 reacts rapidly through addition of $O_2$ to the ortho (C2) to produce peroxy radicals R1-2OO, The
$O_2$ addition to para position (C4) leading to R1-4OO peroxy radicals (not shown in Scheme 1) was found to be
endothermic, therefore negligible (Wang, 2015). Radicals R1-2OO undergo intramolecular H-shifts or ring closures to
form a stable bicyclic intermediate R1-26OO (red). Similarly, R2 reacts rapidly with $O_2$ to form peroxy radical R2-1OO
intermediate, which itself undergoes intramolecular H-shifts or ring closures to form a stable bicyclic intermediate R2-
13OO (red). R1-26OO and R2-13OO intermediates undergo further reactions leading to ring-opening products as shown
in scheme 3 below. 2-Hydroxybenzyl alcohol was proposed by Wang (2015) to form through the reaction of R2 with $O_2$
involving rapid direct H-abstraction. A possible formation pathway of phenol is decomposition of the peroxy radicals R1-
2OO and R2-1OO through $CH_2OH$ radical elimination (Bernard et al., 2013). $CH_2OH$ radical reacts rapidly with $O_2$ to
produce formaldehyde. Catechol was proposed to originate from the reaction of OH radicals with phenol (Atkinson et al.,
1992) and with 2-hydroxybenzyl alcohol (Bernard et al., 20013).

NACs observed in this study (Table 7) are expected to be formed through reaction of OH radicals with BnOH in the

presence of $NO_2$. Scheme 2 briefly summarizes the main mechanistic pathways leading to BnOH NACs, which follow
similar chemistry as those reported for toluene, benzene, and xylenes (Jenkin et al., 2003; Vidovic et al., 2018) and
summarized by Wang et al., (2019). The steps shown in scheme 2 are multi-steps and the reader should consult the
reference papers above for more in depth information. NACs are proposed to originate from secondary reactions of
catechol, 1,3,5-trihydroxy benzene, and 4-hydroxybenzyl alcohol with OH radicals in the presence of $NO_2$ (scheme 2).
These intermediates are proposed to be originated from R1, R2, and R3 adducts. Additional pathways could be initiated
via less well understood aqueous-phase nitration (Kroflic et al., 2018). 4-Nitrocatechol is proposed to be initiated through
the reaction of catechol with OH radicals in the presence of NOx (Finewax et al., 2018). 4-Hydroxy-2-nitrobenzyl alcohol
is proposed to be likely originated from the alkyldihydroxy-para-adduct formed from the OH addition to para position





(scheme 2). 2-Nitrophloroglucinol and 3,4-dihydroxy-5-nitrobenzyl alcohol follow similar reactions involving R3 adduct,
OH radicals, and NO$_2$.

**Scheme 2.** Proposed mechanism for selected NAC species observed in this study.





HOCs were detected in the PP from the oxidation of BnOH in both low and high NOx systems (Table 6). Mechanistic
pathways based on theoretical studies leading to several HOCs (e. g. HOMs) from the atmospheric oxidation of biogenic
and aromatic hydrocarbons have been reported recently in the literature involving unimolecular reaction through
autoxidation, and peroxy and/or alkoxy radical isomerization (Wang, 2015; Jaoui et al., 2021; Piletic and Kleindienst,
2022). The formation of selected HOCs observed in this study is consistent with the following pathways proposed in
scheme 3 involving R2-1300 radical as the starting material. R1-26OO adduct undergoes similar reactions leading to
butenedial and 2,3-epoxy-butanedial as shown in scheme S1 (SI). Tartaric acid, 2,3,5-trihydroxy-4-oxo-pentanoic acid,
and pentaric acid, observed in this study for the first time, are proposed to be initiated by the oxidation of butenedial/2,3-
epoxy-butanedial, 5-hydroxy-4-oxo-2-penatenal, and 4-hydroxy-2,3-epoxypentandial

**Scheme 3.** Proposed mechanism for selected highly oxygenated compounds observed in this study.






(scheme 3). The mechanism leading to butenedial/2,3-epoxybutanal, 5-hydroxy-4-oxo-2-penatanal (scheme S1) was reported by Wand (2015) from the OH oxidation of BnOH, therefore is not shown in scheme 3. Tartaric acid present in the PP at high level (Figure 4), is proposed to be formed through the oxidation of butenedial and/or 2,3-epoxybutanal through classical oxidation of aldehydes and alkenes to carboxylic acid (not shown in scheme 3). Similarly, 2,3,5-trihydroxy-4-oxo-pentanoic acid and pentaric acid are proposed to rise from the oxidation of 5-hydroxy-4-oxo-2-pentenal, and 4-hydroxy-2,3-epoxypentanedial, respectively following similar mechanistic pathways reported by Jaoui et al. (2021) for the formation of methyltartaric acid from 4-hydroxy-2-methyl-but-2-enal involving peroxy and alkoxy radical isomerization (not reported here). In this study, a new mechanism is proposed in scheme 3 leading to the formation of 4-hydroxy-2,3-epoxypentanedial, which is the starting material for pentaric acid formation. It involves several intermediate steps including unimolecular H migration (e.g., 1,5-H shift), ring opening and decomposition. Formaldehyde and glyoxal observed in this study are also shown in scheme 3.

## 4. Summary

In the present manuscript, laboratory experiments were conducted to investigate SOA formation from the oxidation of benzyl alcohol in the presence and absence of NO$_x$. Chamber aerosol collected under these conditions has been analyzed for organic mass to organic carbon ratio, and aerosol yield. In addition, the chemical composition of the gas phase and SOA was analyzed using derivative-based methods followed by gas chromatography-mass spectrometry and high-performance liquid chromatography analysis of the derivative compounds. More than 50 oxygenated organic compounds in the gas and particle phases were identified. While a detailed non-targeted analysis has been made, our primary focus has been to examine highly oxygenated and nitroaromatic compounds. The major components include ring-opening products with high oxygen to carbon ratio (e. g. malic acid, tartaric acid, arabic acid, 2,3,5-trihydroxy-4-oxo-pentanoic acid, and pentaric acid) and ring-retaining products (e. g. benzaldehyde, benzoic acid, catechol, 3-nitrobenzyl alcohol, 4-nitrocatechol, 2-hydroxy-5-nitrobenzyl alcohol, 2-nitrophloroglucidol, 5-(hydroxymethyl)- 3-nitro-1,2-benzyl diol). The presence of some of these products in the gas and particle phases simultaneously provides evidence of their gas/particle partitioning. These oxygenated oxidation products made dominant contributions to the SOA particle composition in both low and high NOx systems. Yields, organic mass to organic carbon ratio, and proposed reaction schemes for selected compounds are provided.

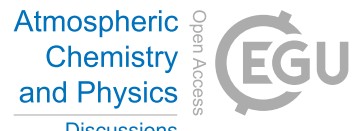

Finally, a set of reaction pathways are proposed that accounts for selected reaction products observed in this study
from BnOH photooxidation in the presence of OH radicals, including NACs and HOCs. The proposed mechanism is
based on (1) theoretical studies reported previously in the literature and (2) mechanisms associated with aromatics
oxidation (e.g., benzene, toluene, xylenes…). New pathways were proposed for the formation of newly observed highly
oxygenated compounds tartaric acid, 2,3,5-trihydroxy-4-oxopentanoic acid, and pentaric acid. Butenedial/2,3epoxy-
butandial, 5-hydroxy-4-oxo-2-pentenal, and 4-hydroxy-2,3-expoxypentanedial were proposed as the starting intermediate
species leading to these highly oxygenated compounds. While theoretical studies involving unimolecular reactions were
developed focusing mainly on ring-containing products (Wang, 2015, Piletic and Kleindienst, 2022), similar theoretical
investigations focusing on linear species (Jaoui et al., 2021) as HOCs reported in this study will help strengthen the
pathways proposed here.
The results of this study potentially have atmospheric implications for areas impacted by benzyl alcohol including
urban and indoor areas and contribute to understanding the formation of ambient SOA from oxygenated anthropogenic
precursors. Nitroaromatics are pollutants of concern due to their toxicity, light-absorption properties, and relatively long
residence times in the environment. HOCs may partition into pre-existing particles or be involved in new particle
formation.

*Data Availability.* The data used in this study can be found at: https://catalog.data. gov/dataset/epa-sciencehub. DOI:
613    10.23719/1527893.


*Competing interests.* The authors declare no competing financial interest.

*Disclaimer.* This work has been subjected to the U.S. Environmental Protection Agency's administrative review and
approved for publication. The views expressed in this article are those of the authors and do not necessarily represent the
views or policies of the U.S. Environmental Protection Agency. Mention of trade names does not constitute endorsement
or recommendation of a commercial product by U.S. EPA.







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





**Table 1.** Initial conditions for BnOH experiments in the presence and absence of NO.

| Exp. IDs | BnOH (ppb) | $H_2O_2$ (ppm) | NO[a] (ppb) | Seed surface area (nm² cm⁻³) | BnOH/NO (ppb/ppb) | T (°C) | RH (%) |
|---|---|---|---|---|---|---|---|
| ER-889 | 385 | - | 178 | $4.67 \times 10^7$ | 2.2 | 24.5 | 31.0 |
| ER-890 | 355 | - | 96 | $4.94 \times 10^7$ | 3.7 | 24.5 | 31.1 |
| ER-891 | 723 | - | 188 | $9.88 \times 10^7$ | 3.8 | 24.6 | 31.3 |
| ER-892 | 319 | 3.04 | - | $1.36 \times 10^6$ | - | 25.7 | < 4.0 |

T: temperature; RH: relative humidity. Seed aerosol at 1 µg m⁻³. [a]: The initial NOx during the irradiations was greater than 98% NO.

**Table 2.** Steady-state GP and reacted BnOH and NO concentration during the irradiations.

| Exp. IDs | NO (ppb) | Reacted NO (ppb) | BnOH (ppb) | Reacted BnOH (ppb) | BnOH/NO ratio (ppb/ppb) | $O_3$ (ppb) | NOy (ppb) |
|---|---|---|---|---|---|---|---|
| ER889 | 78 | 100 | 132 | 253 | 1.7 | 30 | 163 |
| ER890 | 9 | 87 | 132 | 223 | 14.7 | 147 | 80 |
| ER891 | 29 | 159 | 387 | 336 | 13.4 | 11 | 146 |
| ER892 | - | - | 85 | 234 | - | 28 | - |


**Table 3.** Formation and yields of SOA ($Y_{SOA}$) and SOC ($Y_{SOC}$). All organic and carbon aerosol masses are corrected for
a wall loss of 0.067 h⁻¹ (Kleindienst et al., 2012).

| Exp. IDs | SOA (µg/m³) | SOC (µgC/m³) | SOA/SOC | $Y_{SOA}$ (%) | $Y_{SOC}$ (%) |
|---|---|---|---|---|---|
| ER889 | 39.6 | 23.2 | 1.7 | 3.6 | 2.7 |
| ER890 | 56.1 | 30.3 | 1.9 | 5.7 | 4.0 |
| ER891 | 119.5 | 58.9 | 2.0 | 8.1 | 5.1 |
| ER892 | 52.9 | 24.8 | 2.1 | 5.2 | 3.1 |






**Table 4.** Steady state carbonyl concentrations (ppmV) during BnOH oxidation (FH: formaldehyde; AH: acetaldehyde;
Ac: acetone; MA: methacrolein; BN: 2-butanone; BnAld: benzaldehyde; G: glyoxal; MG: methylglyoxal).

| Exp. ID | FH | AH | Ac | MA | BN | BnAld | G | MG |
|---------|-------|------|-----|-----|-----|-------|-----|-----|
| ER889 | 2.4 | 1.2 | 1.0 | 1.0 | 0.6 | 23.09 | 5.0 | 0.6 |
| ER890 | 1.5 | 2.8 | - | - | 2.9 | 18.2 | 3.8 | 0.4 |
| ER891 | 5.1 | 2.5 | 1.3 | 2.0 | 1.4 | 30.8 | 8.6 | 0.6 |
| ER892 | 181.7 | 23.5 | - | 0.8 | 0.8 | 5.2 | 7.8 | 1.6 |

























**Table 5.** Summary of selected reaction products detected and identified either in gas-phase (GP), particle phase (PP) or
both from BnOH/NOx, and BnOH/H$_2$O$_2$ experiments. Tables 6 and 7 shows additional aerosol species with high oxygen
to carbon ratio and/or nitro group. NA: not applicable. [a]: underivatized m/z are given. *: identified with authentic standard.

| IUPAC/common nomenclature | Formula | m/z BSTFA (EI) | MW [MW$_{BSTFA}$] (g mol$^{-1}$) | Proposed Structure | Detected |
|---|---|---|---|---|---|
| Benzyl alcohol (BnOH) | C$_7$H$_8$O | 165, 91, 135, 180, 73 | 108 [180] | | GP |
| Phenol | C$_6$H$_6$O | 73, 151, 166, 94, 65 | 94 (166) | | GP, PP |
| Benzaldehyde (BnAld) | C$_7$H$_6$O | 106, 105, 77, 77, 51 | 106 (NA) | | GP, PP |
| Benzene-1,2-diol (catechol) | C$_6$H$_6$O$_2$ | 239, 255, 80, 283, 73 | 110 (254) | | PP |
| Benzoic acid | C$_7$H$_6$O$_2$ | 179, 105, 135, 77, 194 | 122 (194) | | GP, PP |
| Salicylaldehdye | C$_7$H$_6$O$_2$ | 179, 105, 135, 77, 194 | 122 (194) | | GP |
| 3-Hydroxy benzaldehyde | C$_7$H$_6$O$_2$ | 179, 105, 135, 77, 194 | 122 (194) | | GP |
| 2-Hydroxybenzyl alcohol (salicyl alcohol) | C$_7$H$_8$O$_2$ | 73, 253, 179, 268, 147 | 124 (268) | | GP, PP |
| 4-Hydroxybenzyl alcohol | C$_7$H$_8$O$_2$ | 73, 179, 253, 268, 147 | 124 (268) | | GP, PP |
| 4-Hydroxybenzoic acid (*p*-salicylic acid) | C$_7$H$_6$O$_3$ | 267, 223, 193, 282, 73 | 138 (282) | | PP [H$_2$O$_2$] |




**Table 6.** Highly oxygenated products (O:C > 1.3) identified in benzyl alcohol photooxidation in the presence of NOx, or H₂O₂. *: identified with authentic standard. *L*-Tartaric acid and *D*-tartaric acid co-elute. The structure of 4-oxo-D-arabonic acid isomer and 2,3,5-Trihydroxy-4-oxopentanal isomer are shown for trihydroxy-oxo-pentanoic acid, and trihydroxy-oxo-pentanal, respectively. Four peaks with similar fragments/adducts as pentaric acid were observed.

| Nomenclature | Chemical Formulae | O/C Ratio (by wt) | *m/z* BSTFA Derivative (CI-CH4); (EI) | MW (MW$_{BSTFA}$) | Proposed Structure |
|---|---|---|---|---|---|
| Epoxysuccinic acid (2 peaks) | $C_4H_4O_5$ | 1.7 | 187, 261, 73, 277, 173 <br> 73, 173, 261, 129, 143 | 132 (276) | |
| 2-Hydroxybutanedioic acid* (malic acid) | $C_4H_6O_5$ | 1.7 | 233, 335, 73, 307, 351 <br> 73, 147, 233, 245, 335 | 134 (350) | |
| Trihydroxy-oxo-pentanal (5 peaks) | $C_5H_8O_5$ | 1.3 | 73, 275, 203, 349, 393 <br> 147, 73, 349, 233, 259 | 148 (364) | |
| *meso*-Tartaric acid* | $C_4H_6O_6$ | 2.0 | 423, 321, 277, 439, 73 <br> 73, 147, 292, 219, 423 | 150 (438) | |
| *L*-Tartaric acid* | $C_4H_6O_6$ | 2.0 | 423, 321, 277, 439, 73 <br> 73, 147, 292, 219, 423 | 150 (438) | |
| Trihydroxy-oxo-pentanoic acid (8 peaks) | $C_5H_8O_6$ | 1.6 | 73, 437, 363, 481, 493 <br> 217, 73, 147, 437, 292 | 164 (452) | |
| *D*-Arabinonic acid* (Arabic acid) | $C_5H_{10}O_6$ | 1.6 | 361, 217, 73, 435, 525 <br> 204, 437, 73, 147, 319 | 166 (526) | |
| Pentaric acid* (4 peaks) | $C_5H_8O_7$ | 1.9 | 525, 333, 407, 435, 73 <br> 73, 292, 189, 407, 525 | 180 (540) | |





**Table 7.** NACs identified in benzyl alcohol photooxidation in the presence of NOx.

| Nomenclature | Chemical Formula Rt (min) | *m/z* BSTFA Derivative (CH₄-CI) (EI) | MW (MW$_{bstfa}$) | Observed in GP; PP [GP/PP ratio] | Proposed Structure |
|---|---|---|---|---|---|
| 3-nitrobenzyl alcohol[a] | C₇H₇NO₃ (25.93) | 226, 210, 180, 136, 73<br>210, 180, 165, 194, 73 | 153 (225) | PP, GP [1.71] |  |
| 4-nitrocatechol[a] | C₆H₅NO₄ (30.86) | 300, 284, 328, 254, 73<br>73, 284, 299, 269, 223 | 155 (299) | PP, GP [0.08] | |
| 2-hydroxy-5-nitro benzyl alcohol[a] (4 isomers) | C₇H₇NO₄ (34.26) | 314, 298, 268, 342, 73<br>298, 283, 191, 314, 73 | 169 (313) | PP, GP [0.08] | |
| 2-nitro phloroglucinol[a] (4 isomers)[b] | C₆H₅NO₅ (35.62) | 388, 372, 416, 428, 73<br>73, 372, 387, 284, 306 | 171 (387) | PP | |
| 3,4-dihydroxy-5-nitrobenzyl alcohol (4 isomers)[c] | C₇H₇NO₅ (38.18) | 388, 372, 416, 428, 73<br>73, 224, 3876, 401, 356 | 185 (401) | PP | |

[a]: identified using authentic standards. [b]: Three additional peaks eluted at 33.76, 34.70, 34.76 min with similar mass spectra as those recorded for 2-nitrophloroglucinol standard were detetcted, and the structure given here is for 2-nitrophloroglucinol. [c]: Three additional peaks eluted at 35.94, 36.60, 38.18 min with similar mass spectra were detetcted.







**Figure 1.** Portion of GC-MS total ion chromatogram in EI mode of GP underivatized denuder extract (a) ER-889 (presence of NOx), (b) ER-892 (absence of NOx), and (c) ER889- (presence of Nox) as BSTFA derivatives.





F1: BnOH/NOx     E1: methanol extract     F2: BnOH/H2O2     E2: methanol extract


**Figure 2.** Portion of GC-MS total ion chromatograms (EI mode) of particle-phase extracts: (top) BSTFA derivatized sample form ER-889 (presence of NOx), (middle) BSTFA derivatives from ER-892 (absence of NOx), (bottom) effect of mixture changes in filter and methanol extract appearance: BnOH/NOx filter (F1); BnOH/H$_2$O$_2$ (F2). The same volume of air was sampled on each filter.




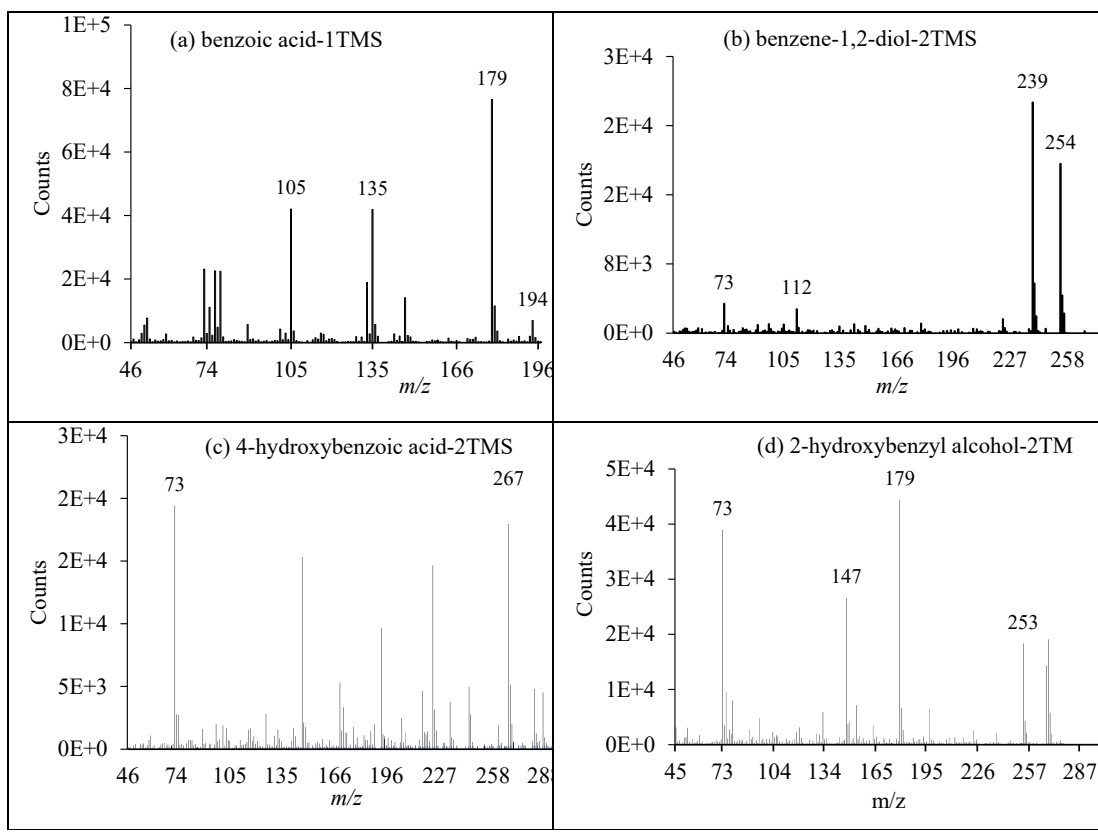

**Figure 3.** Positive EI mass spectra of BSTFA derivatives of selected ring-containing products: benzoic acid, benzene-1,2-diol, 4-hydroxybenzoic acid; and 2-hydroxybenzyl alcohol.

**Figure 4.** Portion (25-34 min) of GC-MS extracted ion chromatograms (CI-CH$_4$) at *m/z* 423 (green); m/z 437 (blue); and m/z 525 (red)
merged in one chromatogram (a) BnOH in the presence of NOx; (b) BnOH in the presence of H$_2$O$_2$ and absence of NOx; (c) Chamber
background. Red and top blue: right axis.



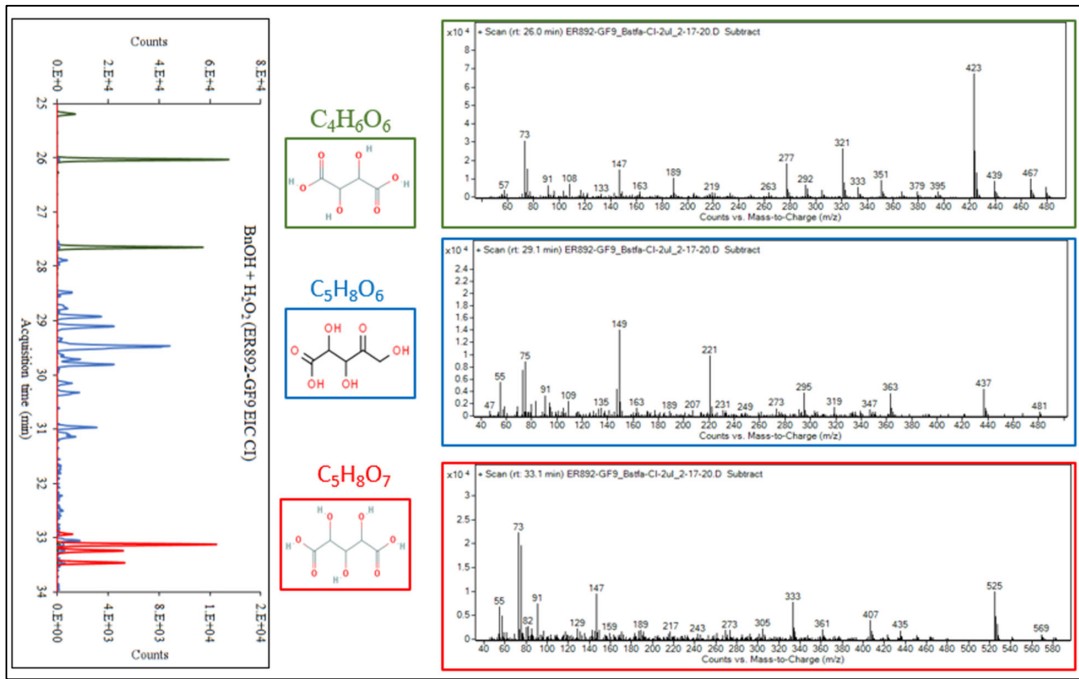

**Figure 5.** Mass spectra (methane-CI) of ester TMS derivatives of meso-tartaric acid (top right), trihydroxy-oxo-pentanoic acid (middle right), (c) pentaric acid (bottom right), along with the portion of GC-MS extracted ion chromatograms shown in figure 6. Chemical formulae and chemical structure associated with each group is given in the middle column.



**Figure 6.** Molecular structures of pentaric acid and its isomers (a); mass spectra of TMS derivatives of pentaric acid acquired for smog chamber SOA (EI: b, CI: c) and authentic standard (d: EI); Major pentaric acid fragments observed in EI mode (e).





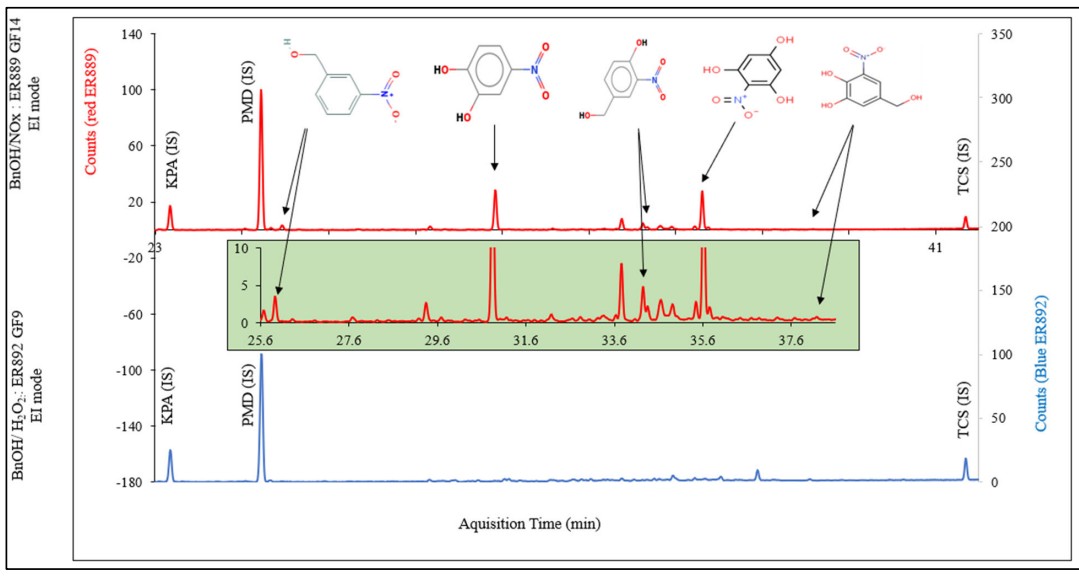

**Figure 7.** Portion of GC-MS extracted ion chromatograms (EI mode) at *m/z* 210, 165 (IS), 299 (IS), 300, 298, 372, 388 associated with
nitroaromatic compounds merged in one chromatogram (red) BnOH in the presence of NOx (ER889); (blue) BnOH in the presence of
$H_2O_2$ and absence of NOx (ER892).



