# Peer review of "Yields and molecular composition of gas phase and secondary organic aerosol from the photooxidation of the volatile consumer product benzyl alcohol: formation of highly oxygenated and hydroxy nitroaromatic compounds"

_Atmospheric Chemistry and Physics, 2022_

## Author Response (AR2)

**UNITED STATES ENVIRONMENTAL PROTECTION AGENCY**
**Center for Environmental Measurements & Modeling**
**Research Triangle Park, NC  27711**

March 08, 2023

========================================================================
**Editor decision: Publish subject to minor revisions (review by editor)**
by Barbara Ervens
**Public justification (visible to the public if the article is accepted and published)**:
Dear Authors,

many thanks for responding thoroughly to the reviewer comments. I noticed that you did only very minor changes to the manuscript. Please note that reviewer #1 (R1) asked for major revisions of your manuscript. Therefore, please address their comments also in the manuscript by adding information mentioned in your response to the manuscript, e.g. in response to R1 comments 2, 3, 5 - 7, and R2 comments 3 - 11. Please also make clear in your next response where in the text additional information has been added (line and/or section numbers) or why you did not add text in response to the reviewer comments.
After this has been satisfactorily addressed, I will be happy to accept your paper for publication in ACP.
Sincerely,
Barbara Ervens
27 Jan 2023

========================================================================
Our response here is focused mainly on the editor comments shown above. Please see our previous response to reviewers comments at the end of this file.

**\*Comments by the reviewers are in blueprint, answers by the authors are in normal print.**

**Reviewer # 1.**

**1.**  The study by Wang (2015) is applicable for atmospheric like conditions of reagents. This is not the case for the conditions in the smog chamber of this study where hundreds of ppbv of reagents are used. To justify the mechanism shown in Scheme 1 it is necessary to have a better understanding on the concentration of NO and $NO_2$ during the different experiments. NO can be sort of extrapolated from table 2 though it is less clear what the concentration of $NO_2$ The table lists NOy values but is that really NOy or is it NOx? I think the value of $NO_2$ is important as in the study by Wang (2015) it is stated that for

high NO$_2$ (100 ppbv) the reaction of R1 (from Scheme 1) with NO$_2$ can compete with the reaction with O$_2$. If this is true, then the mechanisms proposed would only be part of the chemistry and the pathways following the reaction with NO$_2$ should be included.

**Response.** In addition to our previous response, we added the following to the main manuscript (page 22, lines 548-554).

"As in Wang (2015), the reverse reaction that gives R1 + O$_2$ is certainly competitive with the cyclization. Moreover, this reversible reaction sustains the concentration of R1 with the radical now delocalized within the aromatic structure rather than associated with the substituent O$_2$. By this approach, reactions of R1 lead to the formation of phenol and subsequent formation of catechol as shown in Scheme 1. However, we do not have quantitative product data that would give insight to the R1-2OO and R2-1OO decomposition (reverse) rate with respect to the O$_2$-cyclization rate. Note that the R1 reaction to form the phenol is independent of NO$_2$ whereas the reaction of R1 to form nitrobenzyl alcohol is dependent on NO$_2$."

**2.** What is the rate of decomposition of R1-2OO and R2-1OO (from Scheme 1)? Wang (2015) calculate rates for the ring closure larger then 8x10$^3$ s$^{-1}$ for both isomers. The decomposition would have to be very fast to compete. In their study, (Wang, 2015) suggest that a possible path of formation for phenol and catechol could be the decomposition of R1, not R1-2OO. Although, even for R1 they seem to claim that their theory would mainly predict reaction with O$_2$ for R1.

**Response.** In addition to our previous response, we added the following section to the main manuscript (page 23, lines 569-582).

"According to Wang (2015) calculation, at high NO$_2$ (100 ppbv) the reaction of R1 with NO$_2$ can compete to a minor degree with the reaction with O$_2$, therefore R1 possibly forms minor amounts of nitrobenzyl alcohol. This is similar to that of other alkylbenzenes, such as toluene, although at much higher NO$_2$ concentrations (Wang, 2015). According to the Wang calculation at 100 ppb NO$_2$, up to 30% of the R1 radical can form nitrobenzyl alcohol (unspecified nitro location on the ring). When including the R2 radical which forms negligible levels of nitrobenzyl alcohol, no more than 20% (and likely much less under our conditions) of the initial ring retaining radical would form nitrobenzyl alcohol. As noted in the caption to table 1, the initial NOx is 98% NO and thus there is an extremely low initial concentration of NO$_2$. As the extent of reaction increases, the NOy concentration increases as do other more oxidized forms of NOx. Thus, the addition of NO$_2$ (~100 ppb) to the carbon-centered radical on the ring is of the same order as that described by Wang (2015) as being of minor importance compared to O$_2$ addition.

Thus, even though NOy concentrations in these experiments are higher than ambient concentrations, we consider the findings and mechanisms of Wang (2015) to be relevant to this work. We only have an upper limit to the $NO_2$ concentration, that is, NOy-NO. NOy represents in the NOx-NO from the oxides of nitrogen (NOx) monitor. It is well known that the NOx monitor in the NOx-NO channel measures other oxidized organic compounds in addition to $NO_2$."

**3.** Which data/SARs are used for the mechanisms proposed for the reactions of R1-26OO and R2-1300? I understand it is derived based on similarities with other species, but it would be good to give a little bit more details about how it is derive and if this is one possible mechanisms and other paths could also contribute. Also, I assume $NO_2$ in scheme 3 should be NO?

**Response.** In addition to our previous response, we added the following to the main manuscript (page 26, lines 610-616).

"The data used to support the secondary reactions of R2-13OO is primarily product data shown in Scheme 3 which shows fragmentation products from R2-13OO. These products have been well known for alkylbenzenes for many years. We have used analogous pathways for these alkylbenzenes to provide an understanding as to how observed fragmentation products of BnOH are formed. However, it is important to note that these fragmentation pathways follow from the formation of alkoxy radicals and not peroxy radicals. Thus, NO is required in the system. (Note: $RO_2 + RO_2 \rightarrow 2RO + O_2$ is too slow to provide the alkoxy radicals needed for the fragmentation reaction to occur.) Similar pathways are also applicable for R1-26OO."

**5.** Page 9, line 220: since the NO is high so that $RO_2+RO_2$ reactions are minimized, why is in scheme 3 indicated that $RO_2$ could be a reaction partner?

**Response.** Of course, the RO2 + RO2 reaction is small when moderate to high concentrations of NOx are present. **We included the reaction for completeness.**

**6.** Page 10, line 224: how is BnOH going to react with NOx?

**Response.** We believe the reviewer refers to page 10, line 254. The BnOH reaction with NO will occur when the reaction of RO2 with NO is competitive to ring closure. **Regardless, according to Wang (2015) the reaction of NO2 with R1 will occur only to a minor degree compared with ring closure**. These have already been covered in the text. However, to reflect the editor concern we added the following

sentences to the revised manuscript (page 10, lines 263-365):

"The BnOH reaction with NO will occur when the reaction of $RO_2$ with NO is competitive to ring closure.

Regardless, according to Wang (2015) the reaction of $NO_2$ with R1 (Scheme 1) will occur only to a minor degree

compared with ring closure."

**7.** Page 10, lines 225-226: is there a way to separate the yield of SOA from the introduce concentration of SOA? As the yield should be used in models and it would need to be independent on the concentration of SOA, I assume?

**Response.** While inorganic aerosol seed is added to the chamber, SOA is only formed from the chemical reactions taking place. Several SOA products are shown in scheme 2. The measured SOA mass is divided by the reacted BnOH is used to obtain the yield. As noted by the reviewer, the measured yield of SOA from BnOH can be used in air quality models.

**Reviewer # 2**

Specific Comments

**3.** Lines 135: How these samples were collected?

**Response.** This was corrected as suggested by the reviewer. The following sentence s were added to the revised manuscript (page 6, line 135-137):

"Air samples were drawn for 20 min at a rate of 0.50 L min−1 through an impinger containing 5 mL of a DNPH

solution in acetonitrile. The resulting solutions were analyzed by high-performance liquid chromatography with a

ultraviolet detector (HPLC/UV) (Smith et al., 1989)."

3. Line 142: How did you correct the formaldehyde values for interference from the NO2-DNPH product?

**Response.** The LC chromatographic program is such that early peaks are separated. Also, DNPH is present in excess, so it is not the limiting reagent and does not interfere with formaldehyde and other products. Thus, NO2-DNPH is not an issue. Additional information can be found in the reference of Smith et al. 1989 (see reference section) and no additional text is necessary.

4. Lines 192–199: Please discuss the potential effects of gas-wall partitioning of benzyl alcohol and reaction products on your results in light of the modeling study of Krechmer et al. ES&T, 54, 12890, 2020.

**Response.** Our wall losses were experimentally derived. Thus, we do not use the modeling approach described in the reference. This was discussed in the manuscript on page 8, lines 201-206 and we feel that no additional text is necessary.

5. Line 220: What do you mean by "minimized"? Please provide a more quantitative description of the relative contributions of RO2 + NO and RO2 + RO2 reactions.

**Response.** As described in the text with reference to the Wang paper, RO2 radicals preferentially react by ring closure. Thus, the RO2 + NO reaction is generally not competitive in the system. From a modeling study, the RO2 concentration can be obtained. In our experimental study, RO2 levels cannot be determined, and a quantitative representation is difficult to obtain. However, this competition is not important for our schematic interpretation except for Scheme 3 where the fragmentation products are produced from the alkoxy radicals formed.

6. Line 235: Does this wall loss rate account for differences in size distributions in different experiments?

**Response.** The wall loss rate is based on mass and thus the size distribution is only an ancillary aspect to this study. We do not use size distribution in our interpretation. To reflect the editor concern, we added the following sentence to the manuscript on page 9, line 243 and reads:

"The wall loss rate is based on mass and not on the size distribution of the particles."

7. Line 248: What about possible reactions of benzyl alcohol with other species after partitioning to the walls?

**Response.** The study was not designed to measure wall reactions of BnOH. Wall reactions tend to be most relevant for NOy compounds. We feel that this is outside of the objective of this study.

8. Line 285: What are typical derivatization efficiencies?

**Response.** For both DNPH and BSTFA the derivations are greater than 90%.

9. Line 285: What happens to oligomers when they are exposed to derivatizing agents?

**Response.** Oligomers from the aromatic compounds tend not be derivatized by BSTFA. In past studies, this consideration has not been found to be an issue. We have not examined such a possibility in the

BnOH system, but consider it to be of negligible importance and, regardless, it is considered to be outside of the scope of the present study.

**10.** Line 510: What are measured or estimated O3 and NO3 concentrations and how do you know they do not impact product composition?

**Response.** O3 is known to react primarily with compounds having double bonds and certainly not aromatic compounds. O3 concentrations are low and given in Table 2. Likewise, the NO3 reaction with aromatic compounds tends to be very slow and unlikely to have any impact in these systems. Moreover, given the O3 + NO2 reaction which gives NO3 would only react with BnOH to a negligible degree. Of course, in the absence of NOx, there is no NO3 produced. However, to reflect the editor concern, we added the following (page 20, lines 520-523):

"Ozone is known to react primarily with compounds having double bonds and certainly not aromatic compounds.

Ozone concentrations are low and given in Table 2. Likewise, the $NO_3$ reaction with aromatic compounds tends to

be very slow and unlikely to have any impact in these systems. Moreover, given the $O_3 + NO_2$ reaction which gives

$NO_3$ would only react with BnOH to a negligible degree."

**11.** Since nitroaromatics are an important aspect of this study, I suggest reporting NO2 concentrations and discussing how they compare to the atmosphere. Formation of nitroaromatics involves competition between NO2 and O2, so if the NO2 concentrations in these experiments are unrealistically high, then the nitroaromatic products detected here are much less likely to be formed in the atmosphere.

**Response.** In this study, we have not measured NO2, but an upper limit can be determined from NOy – NO. However, we agree that the competition between NO2 and O2 with R1 (and R2) will determine the branching ratio for the two compounds. However, for studies that have been performed with single ring aromatic compounds, an NO2 concentration in the single digit part-per-million range is required for any significant level. The reason we are able to detect the NACs is due to the large volume of air sampled and the sensitivity of the GC-MS.

This comment is related to reviewer #1 R1 and R2 comments. We have addressed this comment in the text added to the manuscript in our response to comments 1 and 2 from reviewer #1.

=========================================================================
**Below is our previous response to the reviewers comments**

**\*** Comments by the reviewers are in blueprint, answers by the authors are in normal print

**1.** The study by Wang (2015) is applicable for atmospheric like conditions of reagents. This is not the case for the conditions in the smog chamber of this study where hundreds of ppbv of reagents are used. To justify the mechanism shown in Scheme 1 it is necessary to have a better understanding on the concentration of NO and $NO_2$ during the different experiments. NO can be sort of extrapolated from table 2 though it is less clear what the concentration of $NO_2$ The table lists $NO_y$ values but is that really $NO_y$ or is it $NO_x$? I think the value of $NO_2$ is important as in the study by Wang (2015) it is stated that for high $NO_2$ (100 ppbv) the reaction of R1 (from Scheme 1) with $NO_2$ can compete with the reaction with $O_2$. If this is true, then the mechanisms proposed would only be part of the chemistry and the pathways following the reaction with $NO_2$ should be included.

**Response.** As noted in the Caption to Table 1, the initial $NO_x$ is 98% NO and thus there is an extremely low initial concentration of $NO_2$. As the extent of reaction increases, the $NO_y$ concentration increases as do other more oxidized forms of $NO_x$. Thus, the addition of $NO_2$ (~100ppb) to the carbon-centered radical on the ring is of the same order as that described by Wang (2015) as being of minor importance compared to $O_2$ addition. Thus, even though $NO_y$ concentrations in these experiments are higher than ambient concentrations, we consider the findings and mechanisms of Wang (2015) to be relevant to this work. We only have an upper limit to the $NO_2$ concentration, that is, $NO_y$-NO. To answer the first query, the quantity $NO_y$ is that represent in the $NO_x$-NO from the oxides of nitrogen ($NO_x$) monitor. It is well known that the $NO_x$ monitor in the $NO_x$-NO channel measures other oxidized organic compounds in addition to $NO_2$. Therefore, $NO_y$ is the correct representation for the final column in Table 2.

According to the Wang (2015) calculation, R1 can compete to a minor degree with $O_2$ for the decentralized R1 radical. As noted by Wang (2015) this is similar to that of other alkylbenzenes, such as toluene, although at much higher $NO_2$ concentrations. According to the Wang calculation at 100 ppb $NO_2$, up to 30% of the R1 radical can form nitrobenzyl alcohol (unspecified nitro location on the ring). When including the R2 radical which forms negligible levels of nitrobenzyl alcohol, no more than 20% (and likely much less under our conditions) of the initial ring retaining radical would form nitrobenzyl alcohol. We do not feel this level of detail is appropriate for Scheme 1 and have thus have not shown it. However, we do include a sentence in the text that presents the possibility that R1 can form minor amounts of nitrobenzyl alcohol.
The following sentence was added to the original manuscript on page 22. And reads:
"According to Wang (2015) calculation, at high $NO_2$ (100 ppbv) the reaction of R1 with $NO_2$ can compete to a minor degree with the reaction with $O_2$, therefore R1 possibly forms minor amounts of nitrobenzyl alcohol."

**2.** What is the rate of decomposition of R1-2OO and R2-1OO (from Scheme 1)? Wang (2015) calculate rates for the ring closure larger then $8x10^3$ $s^{-1}$ for both isomers. The decomposition would have to be very fast to compete. In their study, (Wang, 2015) suggest that a possible path of formation for phenol and catechol could be the decomposition of R1, not R1-2OO. Although, even for R1 they seem to claim that their theory would mainly predict reaction with $O_2$ for R1.

**Response.** This comment is related to Point 1 with respect to the formation of nitrobenzyl alcohol from R1. As noted above, we have placed a sentence in the text that refers to this possibility. With respect to the decomposition rate, we have taken the terminology from the Wang paper in which decomposition refers to the reverse reaction to give R1 + O2. As in Wang, the reverse reaction is certainly competitive with the cyclization. Moreover, this reversible reaction sustains the concentration of R1 with the radical now delocalized within the aromatic structure rather than associated with the substituent $O_2$. By this approach, we have, in fact, have implicitly shown that reactions of R1 leads to the formation of phenol and subsequent formation of catechol as shown in Scheme 1. However, we do not have quantitative product data that would give insight to the R1-2OO and R2-1OO decomposition (reverse) rate with respect to the $O_2$-cyclization rate. We only have the result from Wang noted above. Finally, note that the R1 reaction to form the phenol is independent of $NO_2$ whereas the reaction of R1 to form nitrobenzyl alcohol is dependent on $NO_2$. We thank the reviewer for helping us to clarify this point.

**3.** Which data/SARs are used for the mechanisms proposed for the reactions of R1-26OO and R2-1300? I understand it is derived based on similarities with other species, but it would be good to give a little bit more details about how it is derive and if this is one possible mechanisms and other paths could also contribute. Also, I assume $NO_2$ in scheme 3 should be NO?

**Response.** The data used to support the secondary reactions of R2-13OO is primarily product data shown in Scheme 3 which shows fragmentation products from R2-13OO. These products have been well known for alkylbenzenes for many years. As suggested by the reviewer, we have used analogous pathways for these alkylbenzenes to provide an understanding as to how observed fragmentation products of BnOH are formed. However, it is important to note that these fragmentation pathways follow from the formation of alkoxy radicals and not peroxy radicals. Thus, NO is required in the system. (Note: RO2 + RO2 --> 2RO + O2 is too slow to provide the alkoxy radicals needed for the fragmentation reaction to occur.) Similar pathways are also applicable for R1-26OO. We thank the reviewer for spotting the typo in Scheme 3, that is, NO2 was written in error and should have been NO.

We agree with the reviewer that the mechanism and species need to be performed carefully and have made every effort to do so. We appreciate this comment.

**Minor/technical comments**
**4.** Section 2.5: I think a little bit more details on how experiments are performed would be beneficial. For example, what is the purity of the synthetic air used in the chamber? When are the reagents injected and how long is the waiting before the background is measured?

**Response.** We have added more detail to the experimental procedure where appropriate.

**5.** Page 9, line 220: since the NO is high so that $RO_2$+$RO_2$ reactions are minimized, why is in scheme 3 indicated that $RO_2$ could be a reaction partner?

**Response.** Of course, the RO2 + RO2 reaction is small when moderate to high concentrations of NOx are present. We included the reaction for completeness.

**6.** Page 10, line 224: how is BnOH going to react with NOx?

**Response.** The BnOH reaction with NO will occur when the reaction of RO2 with NO is competitive to ring closure. Regardless, according to Wang (2015) the reaction of NO2 with R1 will occur only to a minor degree compared with ring closure. These have already been covered in the text.

**7.** Page 10, lines 225-226: is there a way to separate the yield of SOA from the introduce concentration of SOA? As the yield should be used in models and it would need to be independent on the concentration of SOA, I assume?

**Response.** While inorganic aerosol seed is added to the chamber, SOA is only formed from the chemical reactions taking place. Several SOA products are shown in scheme 2. The measured SOA mass is divided by the reacted BnOH is used to obtain the yield. As noted by the reviewer, the measured yield of SOA from BnOH can be used in air quality models.

**8.** Page 10, line 260: for consistency, it would be good to use the same expression for the yields, either as a fraction or as a percentage.

**Response.** The representation of the yield in the text is now consistent.

**9.** Page 14, line 373-373: catechol on one line is mentioned to be observed for both conditions of NOx while in the following line it is mentioned only for experiments without NOx?

**Response.** Catechol was observed in this study in the absence and presence of NOx. This was corrected accordingly. According to Wang, the formation of catechol occurs by the same mechanism in the absence and presence of NOx. The NO2 reaction with catechol then forms nitrocatechol. Thus, NOx is not required to form catechol but required to form nitrocatechol.

The Wang (2015) reference has been noted and is included in the list of citations.

**Reviewer # 2**

Specific Comments

**1.** Line 100: Stating that you used the highest purity compound available is not useful for assessing possible contaminants. Please state the purity.

**Response.** The purity of the reactants is now stated.

**2.** Lines 123, 181, 196: "Equilibrium" should be replaced by "steady state". They are not the same.

**Response.** This was corrected as suggested by the reviewer.

**3.** Lines 135: How these samples were collected?

**Response.** This was corrected as suggested by the reviewer. The following sentence s were added to the revised manuscript:

"Air samples were drawn for 20 min at a rate of 0.50 L min−1 through an impinger containing 5 mL of a DNPH solution in acetonitrile. The resulting solutions were analyzed by high-performance liquid chromatography with a ultraviolet detector (HPLC/UV) (Smith et al., 1989)."

3. Line 142: How did you correct the formaldehyde values for interference from the NO2-DNPH product?

**Response.** The LC chromatographic program is such that early peaks are separated. Also, DNPH is present in excess, so it is not the limiting reagent. Thus, NO2 is not an issue.

4. Lines 192–199: Please discuss the potential effects of gas-wall partitioning of benzyl alcohol and reaction products on your results in light of the modeling study of Krechmer et al. ES&T, 54, 12890, 2020.

**Response.** Our wall losses were experimentally derived. Thus, we do not use the modeling approach described in the reference.

**5.** Line 220: What do you mean by "minimized"? Please provide a more quantitative description of the relative contributions of RO2 + NO and RO2 + RO2 reactions.

**Response.** As described in the text with reference to the Wang paper, RO2 radicals preferentially react by ring closure. Thus, the RO2 + NO reaction is generally not competitive in the system. From a modeling study, the RO2 concentration can be obtained. In our experimental study, RO2 levels cannot be determined, and a quantitative representation is difficult to obtain. However, this competition is not important for our schematic interpretation except for Scheme 3 where the fragmentation products are produced from the alkoxy radicals formed.

**6.** Line 235: Does this wall loss rate account for differences in size distributions in different experiments?

**Response.** The wall loss rate is based on mass and thus the size distribution is only an ancillary aspect to this study. We do not use size distribution in our interpretation.

7. Line 248: What about possible reactions of benzyl alcohol with other species after partitioning to the walls?

**Response.** The study was not designed to measure wall reactions of BnOH. Wall reactions tend to be most relevant for NOy compounds.

8. Line 285: What are typical derivatization efficiencies?

**Response.** For both DNPH and BSTFA the derivations are greater than 90%.

9. Line 285: What happens to oligomers when they are exposed to derivatizing agents?

**Response.** Oligomers from the aromatic compounds tend not be derivatized by BSTFA. In past studies, this consideration has not been found to be an issue. We have not examined such a possibility in the BnOH system, but consider it to be of negligible importance and, regardless, it is considered to be outside of the scope of the present study.

10. Line 510: What are measured or estimated O3 and NO3 concentrations and how do you know they do not impact product composition?

**Response.** O3 is known to react primarily with compounds having double bonds and certainly not aromatic compounds. O3 concentrations are low and given in Table 2. Likewise, the NO3 reaction with aromatic compounds tends to be very slow and unlikely to have any impact in these systems. Moreover, given the O3 + NO2 reaction which gives NO3 would only react with BnOH to a negligible degree. Of course, in the absence of NOx, there is no NO3 produced.

11. Since nitroaromatics are an important aspect of this study, I suggest reporting NO2 concentrations and discussing how they compare to the atmosphere. Formation of nitroaromatics involves competition between NO2 and O2, so if the NO2 concentrations in these experiments are unrealistically high, then the nitroaromatic products detected here are much less likely to be formed in the atmosphere.

**Response.** In this study, we have not measured NO2, but an upper limit can be determined from NOy - NO. However, we agree that the competition between NO2 and O2 with R1 (and R2) will determine the branching ratio for the two compounds. However, for studies that have been performed with single ring aromatic compounds, an NO2 concentration in the single digit part-per-million range is required for any significant level. The reason we are able to detect the NACs is due to the large volume of air sampled and the sensitivity of the GC-MS.